# Australian Sheep Producers’ Knowledge of and Attitude towards Post-Harvest Feedback: A Mixed-Methods Case Study

**DOI:** 10.3390/foods12234264

**Published:** 2023-11-25

**Authors:** Kate Wingett, Robyn Alders

**Affiliations:** 1Faculty of Medicine and Health, University of Sydney, Camperdown 2006, Australia; 2Global Health Programme, Chatham House, London SW1Y 4LE, UK; robyn.alders@anu.edu.au; 3Development Policy Centre, Australian National University, Canberra 0200, Australia

**Keywords:** sheep, producer, Australia, feedback

## Abstract

Reducing food loss is a key target for Sustainable Development Goal 12—responsible consumption and production. This study aimed to explore Australian sheep producers’ knowledge of and attitude towards post-harvest feedback and how these influence pre-consumer losses in the Australian sheep meat value chain. A mixed-methods approach was taken, using a structured survey and focus group discussions. Descriptive analysis was performed on the completed structured surveys and framework analysis was performed on the focus group discussion transcripts. The structured survey results showed that sale method was the main factor influencing the quality of feedback received on carcases. No respondents reported receiving feedback on edible offal. Most producers indicated that they wanted more detailed feedback on carcases and all respondents wanted more detailed feedback on edible offal. Four themes emerged from the focus group discussions: situation, knowledge, and attitudes of producers to the feedback system; factors, enablers, and barriers in the feedback system; equity in the feedback system; and sustainability of the value chain. In addition, several short-to-medium- and long-term strategies were identified by the authors to reduce pre-consumer losses in the Australian sheep meat value chain, based on the results of this study.

## 1. Introduction

Food loss and waste undermines the sustainability of our food systems [1]. Reducing pre-consumer food losses may help increase food and nutrient availability and, in this way, help support food and nutrition security. In addition, reducing food losses minimises the natural resources used for production on a yield basis, particularly in livestock value chains [2].

The majority of losses from Australian livestock supply chains occur at the processing stage [3]. These losses arise from either animal health issues, contamination during processing making the products not fit for human consumption [4], or market drivers [5]. Feedback systems in the value chain are one tool that can help reduce pre-consumers losses, i.e., losses incurred on the farm, at the saleyard/abattoir/manufacturers/retailers or during transport between these places [6,7].

Australia is the second largest producer of sheep meat in the world [8]. In 2021, the Australian sheep meat value chain produced 656,750 tonnes of carcase meat and 84,064 tonnes of offal for human consumption [8]. The industry is export-orientated, and in financial year 2022, there were more than 31,000 businesses involved in the sheep industry, producing sheep meat and wool valued at AUD 8.1 billion [8,9,10]. Since 2007, the National Sheep Health Monitoring Project has been in place in Australia, with one of the key objectives of this project being to provide feedback to producers on animal health conditions found during abattoir inspections [7].

This study aimed to explore Australian sheep producers’ knowledge of and attitude towards post-harvest feedback and how these influence pre-consumer food losses in the Australian sheep meat value chain. The Australian sheep meat value chain was chosen due to the social and economic contribution of the value chain within Australia and globally. In addition, the National Sheep Health Monitoring Project had been running for a decade at the time the study commenced. Feedback was defined as “the returning of a part of the output of the system as input, for correction or control purposes” and was based on the Macquarie Dictionary definition [11]. Guy, Brown, and Banks [12] highlighted the critical need for all value chain actors to be involved in the feedback system, including sheep producers, if feedback systems are to be optimised for sustainable production.

Initially, the following questions were developed to begin the exploration into producers’ attitude to feedback:What feedback do Australian sheep producers receive regarding carcase and offal from the abattoirs?Is there any relationship to the feedback received and the enterprise type/management/sale method?Are Australian sheep producers satisfied with the feedback they receive?

## 2. Materials and Methods

The study was conducted in accordance with the Declaration of Helsinki and as per the guidelines in the national statement on ethical conduct in human research [13]. The protocol was approved by the Human Research Ethics Committee of the University of Sydney (HREC 2016/753). A mixed-methods approach was adopted in this study. To begin with, a structured survey was undertaken, followed up with focus group discussions (Figure 1).

### 2.1. Structured Survey

#### 2.1.1. Study Design and Setting

The structured survey was developed based on the methodology in the American Association for Public Opinion Research [14] by de Vaus [15] and Crawford [16]. The population of interest was self-identified Australian sheep producers.

#### 2.1.2. Sampling and Recruitment

Web-based convenience sampling was used [17] due to challenges the researchers had with accessing a comprehensive list of Australian sheep producers.

#### 2.1.3. Data Collection

A paper version of the structured survey was pre-tested with sheep producers at a major agricultural show in April 2017. As a result, the number of questions was reduced, and the researchers made all the questions closed; where “other” was an option, respondents had the opportunity to provide more information in a free-text box. In addition, the content of the questions on demographics and enterprise structure and activities was amended. Industry terminology used in the structured survey was consistent with language used by Meat and Livestock Australia, the Australian red meat industry marketing and research body, except for hogget. Hogget was included in the final structured survey as the producers in the pre-test valued receiving information on the three age groups—lamb (meat of young sheep), hogget (meat of yearling sheep), and mutton (meat of adult sheep). All questions, including those in the demographic section, were optional.

An electronic version of the structured survey was developed on Survey Monkey Inc., San Mateo, CA, USA, www.surveymonkey.com (accessed on 11 March 2017). The link to this survey was circulated via newsletters, local networks, and social media of rural organisations (including state farming organisations, Country Women’s Association, Farmers for Climate Action and Livestock Biosecurity Network) from May to September 2017.

The survey was structured in four sections:Section 1: Demographics—age, gender, and farming experience;Section 2: Enterprise information—structure, management, products, sale methods, and end points of products;Section 3: Post-farm-gate: feedback received from processors—feedback frequency, content, and value;Section 4: Household meat and offal consumption and understanding of the comparative nutrient value of these products.

This paper reports on Section 1, Section 2 and Section 3 of the survey. Section 1, Section 2 and Section 3 of the survey consisted predominantly of multiple-choice questions with a small number of free-text questions (postcode, gender, area of production systems, and “other” options for management questions on type of enterprise, vaccines, cast-for-age, and selection of breeding traits).

#### 2.1.4. Data Analysis

Due to the small number of completed returned surveys and the non-probability sampling method, only descriptive analysis was performed on the results of the structured surveys. This analysis was conducted in Microsoft Excel 2013. Categorical frequencies were calculated.

Based on the analysis of the survey results, the researchers then had further questions on how producers receive and utilise feedback, their opinion of the system, and the opportunities producers see for the system in the future. The decision was made to investigate these issues using focus group discussions.

### 2.2. Focus Group Discussions

#### 2.2.1. Study Design and Setting

A qualitative descriptive approach was taken as described by Sandelowski [18] in conducting this part of the study. Focus group discussions as described by Kitzinger [19] were chosen as the research tool to collect the data to investigate these issues.

The population of interest was self-identified sheep producers. Initially, focus group discussions were considered for the four major Australian sheep producing jurisdictions (i.e., New South Wales, Victoria, Western Australia, and South Australia). Due to the variation in traceability and feedback programs in the various jurisdictions, the decision was made to limit the geographic reach of the focus group discussion to include sheep producers from one jurisdiction only to reduce the between-group variation in responses due to the jurisdictional location of the sheep producers. The New South Wales feedback system was reflective of the base-level feedback systems in place nationally at the time [20,21].

The aim was to have homogeneity within the groups regarding availability and producer accessibility to advisory services, with a focus on major sheep-producing regions. It was assumed that geography and the type and magnitude of agricultural production in the regions would directly influence the availability and accessibility to advisory services and that major sheep-producing regions would have optimal access. The authors defined a major sheep-producing region as an animal health region that had 10% or more of the New South Wales sheep and lamb population, according to the national agricultural statistics [22].

#### 2.2.2. Sampling and Recruitment

A non-probability, purposeful sampling method was used to achieve the desired group homogeneity and to form groups in the identified sheep-producing regions of New South Wales [23]. There are 11 animal health regions in NSW [24]. A total of five regions met this criterion, representing 79% of the Australian sheep and lamb population [22]. Each group would ideally have 6–8 participants [25], with a minimum of 5 and maximum of 10 participants. Snowball sampling was then used to form each focus group [26] in the relevant animal health region. A contact was made for each region and this initial contact then invited other participants. The initial contact for each region was an acquaintance of one of the researchers; the remaining participants were not known to the researchers. The contact invited participants (local sheep producers) via email to join a discussion on feedback and ways to improve feedback. The email introduced the lead author as a PhD candidate at the University of Sydney and named her supervisors.

#### 2.2.3. Data Collection

The focus groups were held in person during February–March 2020. Due to the movement restrictions associated with the COVID-19 pandemic, focus group discussions were conducted in only three of the five major sheep regions of New South Wales. Attempts were made for an additional focus group discussion to be held online for the other two sheep regions of New South Wales during the pandemic, but the researchers were unsuccessful in recruiting the minimum numbers of participants as the area experienced multiple lockdown restrictions.

Each focus group was held in a hired community facility. There were no non-participants present at two of the discussions and one non-participant at the third focus group discussion. The discussions ranged in length from 1.2 h to 1.8 h. The lead author facilitated the focus groups; the discussions were based on semi-structured guidance questions (Figure 2). The discussions were audio-recorded.

#### 2.2.4. Data Analysis

Framework analysis, as per Ritchie and Spencer [27] and Gale, Heath, Cameron, Rashid, and Redwood [28] was performed, using a qualitative data management system, NVivo14^©^, Burlington, MA, USA, QSR International. The five steps in framework analysis were undertaken, i.e., familiarisation, identifying a thematic framework, indexing, charting and categorising, and interpreting the data.

The voice recordings were transcribed verbatim by the lead author. Notes were made by the lead author during the transcription process as well as during the group discussions. As there were only three focus groups, the lead author was able to review all the material in detail. Key issues and themes were identified from all three groups and similarities and differences were noted at this stage. Categories and codes were identified, creating the first version of the thematic framework. In the indexing stage, codes were applied to the three transcripts. At the end of indexing each group’s transcript, the codes and categories were refined, and the indexing of the already coded transcripts were revised to reduce variation based on the order of indexing. The codes and categories were then charted with the focus groups, noting the similarities and differences between and within the focus groups on attitudes to feedback. Further interpretation of the data was performed, defining factors that influence feedback and mapping the range and nature of feedback received by New South Wales sheep producers.

## 3. Results

### 3.1. Structured Survey

There were 15 responses in total to the structured survey. Of these, three were fully excluded due to insufficient information being provided in the responses.

#### 3.1.1. Demographics

Most respondents identified as male (8/12), and the remainder identified as female (4/12). Two thirds of the respondents (8/12) were aged between 56 and 70 years, one quarter (3/12) were aged between 41 and 55 years, and one respondent (1/12) was 71 years or older. All respondents (12/12) had more than 21 years’ experience as sheep producers, and nearly half (5/12) had more than 41 years’ experience.

#### 3.1.2. Enterprise Information

The properties of most respondents were in New South Wales (7/12), with the remainder located in Victoria (4/12) and Western Australia (1/12). The land area of the properties ranged from less than 250 hectares (3/12) to more than 3000 hectares (2/12). The number of breeding ewes ranged from less than 500 (3/12) to more than 3000 (2/12).

Regarding sale methods of lambs, ewes, and wethers in the previous 12 months, one quarter of the respondents (3/12) had exclusively sold stock through saleyards and one third of respondents (4/12) had exclusively sold stock over the hooks. According to the Meat and Livestock Australia glossary [29], over the hooks is defined as “marketing of cattle/sheep/lambs directly from the farm to an abattoir where a producer is paid for the value of the carcase based on a sliding grid. The skin is also evaluated for length and quality and is purchased by the processor. The seller generally pays for the animal’s transport from the farm to the abattoir. The grazier generally gets paid within a 7-to-14-day period”. Two respondents had used a combination of these two sale methods. The two enterprises that identified as studs used over-the-hook sale methods and paddock sales. The remaining respondents (2/12) reported using a mix of saleyard and direct-to-retail sales.

#### 3.1.3. Post-Farm-Gate

Only one of the respondents (1/12) reported being aware of the end point of any of the products that were produced from their stock. This producer sold lambs and ewes over the hooks and ewes via paddock sales. The producer was aware of the end point of lamb and ewe carcases but not of offal derived from these animals.

Respondents were asked regarding the frequency of feedback received from processors in the previous 12 months. The categories were divided by product type (carcase or offal), age (lamb, hogget, or mutton for carcases and lamb or adult sheep for offal), and breed (merino or other). There was a much higher response rate to the carcase section of this question (11, 10, and 12 for lamb, hogget, and mutton, respectively) compared with the offal section (5 for lamb and 6 for adult sheep).

Fifty five percent of the respondents received feedback every time on lamb carcases (6/11); all these respondents sold lambs exclusively over the hooks. Half of the respondents received feedback every time on hogget (5/10), again receiving feedback when selling over the hook. Feedback on mutton carcases was received every time for 50% of the respondents (6/12), linked to selling over the hook, direct to retailer, or through paddock sales (selling direct from paddock to buyer).

All respondents selling exclusively through saleyards (*n* = 3) reported never receiving feedback on carcases from processors in the past 12 months. One producer reported receiving feedback less than half of the time and sold via saleyards and over the hooks.

The type of feedback received by those selling over the hooks or direct to retailer included carcase weight (7/7), body condition score (6/7), number of carcases condemns (4/7), number of carcases trimmed (2/7), and other (1/7).

Regarding offal, no respondents reported receiving feedback on lamb or adult sheep offal from processors. One producer reported not producing either of these products and the remaining producers reported never receiving feedback on offal (4/5 lamb and 5/6 adult sheep).

Producers were then asked whether they would like more detailed feedback on the above products. The response rate was again greater for carcases (12, 8, and 11 for lamb, hogget, and mutton, respectively) than offal (7 for both lamb and adult sheep). One response was disregarded as the response to this question did not match the response on feedback frequency and production. Most producers indicated they wanted more detailed feedback on carcases (9/11, 6/7, and 8/10). All respondents (7/7) wanted more detailed feedback on both lamb and adult sheep offal.

### 3.2. Focus Group Discussions

There were three focus groups held with New South Wales sheep producers during February and March 2022, with a total of 21 participants (6, 7, and 8 participants in each of the groups, respectively). There were five female and sixteen male participants across the groups, with each group having or one or two females. Participants ranged in experience as sheep producers from 1 to 40+ years. Enterprise types of participants included commercial and stud, wool and meat, and dual-purpose.

As a result of the framework analysis, there were 39 codes developed, divided into 9 categories (Table 1), encompassing 4 major themes. The codes, categories, and themes were developed using deductive (based on peer-reviewed literature, survey results, and semi-structured discussion questions) and inductive on analysis. Key strategies for increasing the effectiveness of the feedback in reducing pre-consumer losses were able to be identified.

The four themes that emerged from the analysis, relevant to the issues being explored through the focus groups, were the following:Situation, knowledge, and attitudes of producers to the feedback system;Factors, enablers, and barriers in the feedback system;Equity in the feedback system;Sustainability of the value chain.

Each of the four themes is discussed below with representative quotations.

#### 3.2.1. Situation, Knowledge, and Attitudes of Producers to the Feedback System

Participants of two groups acknowledged that they were engaged producers and “pretty much on the ball’ (focus group 2, participant 1). The participants in the third focus group did not explicitly state that they were engaged producers but demonstrated this through showing how they have been early adopters of industry initiatives and technology to improve production outcomes. It became apparent that the enterprise type and/or class of stock influenced the type of feedback these producers were seeking, with participants actively seeking objective information on the performance of breeding stock using individual identification and gathering information on the farm (Figure 3).

All groups confirmed that the method of sale was integral to the standard of feedback received and highlighted the variation in the detail of feedback depending on the abattoir where the animals are processed. There was variation between the groups in how they sold their sheep. Participants in focus group 3 predominantly sold through saleyards. Participants in focus group 1 used a blend of saleyards and over the hook, while participants in focus group 2 predominantly sold over the hook. The choice of sale method was driven by optimising sale prices by going where “the market is really hot” (focus group 1, participant 3).

When selling through the saleyards, the only data received were the sale price for the live sheep, via their agent’s post-sale as a phone call, followed up by an email. All groups appreciated the challenge of providing individual feedback to producers from animals sold through the saleyard.

When sold over the hooks, producers received the weight of each carcase, the carcase fat score, and the number of condemned carcases. This information comes from the agent via email, generally within a few days post-slaughter. The reason for the carcase condemnation or trimming is not routinely included in this email from the agent: “they never, ever tell you whether it was arthritis or what it is” (focus group 2, participant 2).

It was also discussed that when selling over the hooks, including service kills, the processors own all products except the carcase and skins. The participants were aware of the importance of offal and other byproducts to the abattoirs in terms of profit, with the abattoirs “maximizing everything that goes through” (focus group 1, participant 2) to offset the slaughter costs. None of the groups had detailed knowledge of offal specifications or markets and asked questions which remained unanswered, such as “do you know the value of the offal, the offal from a singular sheep to the processor?” (focus group 2, participant 4) and “so do we process any offal; do we make sausages and things like that?” (focus group 1, participant 5).

All groups made it very clear that they did not receive any feedback on offal: “You talk about offal and things, we don’t hear” (focus group 3, participant 2); “nothing on offal” (focus group 2, participant 5); “I don’t think I’ve ever seen any feedback on offal” (focus group 1, participant 4). Several participants in focus group 1 were keen for feedback on the offal weight and any disease conditions. This was then challenged by another participant asking, “once you have sold the sheep, the offal isn’t yours, so why should you get feedback on it?” (focus group 1, participant 3.) This was refuted with “if we can actually produce it and make it a better product and the abattoirs do better and that helps the whole system.” (focus group 1, participant 2.)

In more general terms, all participants were seeking more detail in the feedback on carcase and offal quality, including the reason for full condemnations. There was agreement that receiving the reason for carcase condemnation was “often quite a difficult process” (focus group 1, participant 4). On occasions when this information was obtained, there was a degree of scepticism regarding the accuracy of the information provided: “Nine times out of ten, even when you ask, even if you don’t get to see the paperwork, they tell you it is sheep measles” (focus group 1, participant 2). Current feedback was considered “basic” (focus group 2, participant 6) and “limited” (focus group 3, participant 4), with minimal progress made in the past decade. Participants felt they were “flying blind” (focus group 2, participant 6) and being kept “in the dark” (focus group 3, participant 5). The detailed feedback would be used to facilitate evidence-based management decisions on the farm regarding breeding, feeding, and animal health interventions.

#### 3.2.2. Barriers and Enablers in the Feedback System

Several barriers and enablers to establishing and maintaining a feedback system were raised by all the groups. This included how the system is resourced, including funding. The role of producers, agents, abattoirs, industry bodies, and government were discussed.

Commercial in confidence was seen as “mak(ing) it extremely difficult” (focus group 1, participant 4) for abattoirs to share data on offal production with producers. This was highlighted as an issue for data sharing on lean meat yield using dual X-ray absorptiometry (DEXA) technology with “the abattoirs that have it (being) very reluctant to part with the information that they are getting as they believe it is theirs, they own it” (focus group 2, participant 7). An alternative for providing regional-level animal health data to government agricultural services was proposed to assist with managing diseases that are spread by wild animals.

All groups noted that they had limited opportunities to discuss feedback with other value chain actors and asked the researcher whether she was going to discuss feedback with the abattoirs. It was discussed that developing a common operating picture across value chains would be beneficial to better understand what information each sector wants and what is feasible to capture and share. Focus group 3 raised that several decades ago there was a program to develop such a common operating picture with producers selling on forward contracts over the hook. This was supported by the government, producers, and processors. With a change in the market structure, it became more profitable for producers to sell through the saleyards and many producers, “being their own worst enemies” (participant 1), reneged on their contracts with processors, and this ended the program.

Access to feedback in a timely manner was raised as an enabler for producers. It was recommended in focus group 2 that detailed feedback should be received with the kill sheet, “because that’s when you’re keen to look at it” (participant 6).

The need for a trusted advisor to convert information into intelligence was discussed. Focus group 3 raised that there are “lots of sources of information that none of us are aware of so maybe we say we get no feedback but maybe it is there, but so maybe it is the joining of the people giving the feedback and the people that are looking for the information. Maybe there is a gap in the middle there” (participant 4). The same group discussed that the changes in institutional advisory services in the area over time, both government and industry, had a negative impact on unbiased information sharing; this was also raised by focus group 1. 

Technology was generally seen as an enabler for the feedback system. However, there were some challenges raised in association with incorporating new technology into the feedback system, such as who pays and governance (Figure 4). There were challenges raised regarding the Livestock Data Link, in that it was considered to be not well known, with some participants not having logged on.

#### 3.2.3. Equity in the Feedback System for Stakeholders

A strong theme that came from the discussion groups was equity in feedback systems for all value chain actors. There were several factors discussed in this theme that influenced producers’ knowledge and attitude to equity in the feedback system that mostly arose from participants’ individual experiences.

The New South Wales government’s use of abattoir surveillance for the diagnosis of ovine Johne’s disease in the early 2000s has had long-lasting impact on the attitudes of these New South Wales sheep producers towards abattoir feedback. All three groups raised this historical policy without any question or prompting from the facilitator. Identification and feedback in this situation were described as a “weapon used against producers”. Others “didn’t want any (feed) back” (focus group 2, participant 5) at that time, as they were concerned it would be regarding ovine Johne’s disease.

The significant variation in feedback received based on sale method, abattoir, and jurisdiction was raised by all three groups. One participant of focus group 3 felt that the difference in feedback received between mobs sold directly to the abattoirs or via the saleyards was “in this day and age ridiculous”. Participants in focus group 1 were genuinely curious as to why South Australian producers received significantly greater feedback than New South Wales producers. The big supermarkets were flagged as factors influencing sale method and abattoir choice in Focus group 1.

Collecting and sharing data takes human resourcing and financial support, and “asking the abattoir to do more and more work for no extra money” (focus group 1, participant 4) was raised as a factor to consider when looking for ways to improve the feedback system. The example of the Enhanced Abattoir Surveillance Program, a joint project between the South Australian government and the sheep industry [30], that was funded through producer levies and the government, was then provided as a successfully funded feedback program.

The benefits of the traceability system were discussed in all three groups, with focus groups 1 and 2 seeing the benefits for market access and feedback. Participants in focus group 3 questioned these benefits and sought more evidence on the greater good of the system.

The lack of financial recognition for defect-free carcases when selling through the saleyards or high-quality, offal no matter the sale method, was raised by focus group 3. It was seen as inequitable that good producers are financially supporting other producers in the value chain, due to the distribution of the cost of downgraded products by the abattoirs (Figure 5).

#### 3.2.4. Sustainability of the Value Chain

All groups recognised different economic, social, and environmental factors that directly or indirectly influence the functionality of the feedback system. Focus group 3 discussed the diminishing desire for feedback by producers when prices are strong. Similarly, in focus groups 1 and 2, it was raised that while processors are making sufficient profits on offal and other byproducts, there is little financial drive to provide producers with feedback. This included the non-human food chain being a profitable market for abattoirs, such as diversifying offal to the pet food supply chain (Figure 6).

Social factors raised included animal welfare, human health, and nutrition and the environmental impact of raising ruminants for meat production. It was discussed that animal rights may be a driver for feedback in the future “ethically they may say if an animal is giving its life for you to eat meat, you have to use the whole animal” (focus group 1, participant 2). The role of the medical profession in optimising nutrient availability was questioned: “Where does the medical profession fit in to all of this? We’re talking about low haemaglobin levels, are they just getting the iron in a tablet? Is that the easy way out?” (focus group 1, participant 3). Recommendations from elements within the scientific community and social media influencers to eat less red meat to reduce environmental impacts were raised by focus group 1.One participant in focus group 3 was “much more interested in the feedback on that, than the feedback on profit; on greenhouse gas emissions, including on a whole farm basis, sequestering carbon in the soil and all that stuff”.

## 4. Discussion

The results show that the Australian sheep producers participating in this study would like more detailed feedback about the products derived from their sheep. The focus groups highlighted the lack of feedback on offal quality and quantity from the abattoir to these producers. Among the focus group discussion participants, there was a consensus of stagnation/slow progress about the feedback system and that, despite information being collected, it was not being converted to accessible intelligence at the enterprise, regional, or, for New South Wales at least, jurisdictional level. There was a sense of inequity within the value chain, in that the feedback any producer receives is highly dependent on the sale method and the abattoir and that there appeared to be inequitably financial contributions to the system.

It was discussed that there are potential benefits beyond the farm and abattoir to receiving feedback and reducing pre-consumer losses from the Australian sheep meat value chain. This included the optimisation of export market access for the whole industry, increased nutrient availability for people, more accurate assessment of greenhouse gas emissions, improved animal welfare, and better use of an animal that has been slaughtered for food production.

Proposed solutions included providing information on the reason for condemnation of carcases/carcase meat and offal with or on the kill sheet and using technology to facilitate the delivery of the same feedback to every sheep producer, regardless of the method of sale, abattoir, or jurisdiction. The need for these producers to be better equipped to turn data into intelligence was highlighted. The challenges with funding the system and the legalities of information sharing were raised in the focus group discussions.

With the National Sheep Health Monitoring Project having been in place for more than 15 years, the Australian sheep meat value chain is in an ideal position to continue to improve the feedback system and reduce pre-consumer losses. With mandatory electronic individual-sheep identification being introduced nationally in Australia by 2025 [31], there is an opportunity for more tailored feedback to be delivered to producers and more detailed data analysis to be undertaken at a population level.

In addition to this national project, an enhanced abattoir surveillance program was in place from 2007 to 2021 in South Australia, providing real-time feedback to producers that had directly consigned sheep/lambs to a particular abattoir as well as publishing regional animal health benchmarking reports [30]. A survey conducted by the Department of Primary Industries and Regions South Australia regarding the enhanced abattoir surveillance program then in place in South Australia reported that 96% of producers who had received notification of a disease condition in their sheep/lambs at slaughter valued the feedback. Approximately 67% of producers were unaware of the presence of the disease on their farm prior to receiving the feedback and 40% of producers had implemented management changes based on the information received from the enhanced abattoir surveillance program [32].

Based on the themes from the focus group discussions and the findings from the South Australian Enhanced Abattoir Surveillance Program, the authors have identified several key strategies to achieve this continuous improvement nationally. These strategies vary in terms of the level of intervention required and as such have been considered to be short-to-medium-term and longer-term strategies.

Short-to-medium-term strategies include:Routinely including the reason for full carcase condemnations on the kill sheet;Publishing the number of full carcase condemnations in the national agricultural dataset, as the New Zealand Government does [33];Expanding the current national data analysis and reporting from the National Sheep Health Monitoring Project to increase the frequency (consider quarterly instead of annually) and geographic granularity (consider natural resource management regions as base geographic region instead of jurisdiction).

Longer-term strategies include:Providing feedback to producers regardless of sale method or abattoir. The feasibility of this will increase when individual sheep electronic identification is mandated nationally in 2025 [31]. Consideration must be given to the disproportionate costs to small-to-medium-size abattoirs of setting up and maintaining data collection and dissemination systems [6].Routinely providing quantified feedback on not only carcase condemnations but also offal condemnations. Some of these animal health conditions have little to no effect on sheep productivity, e.g., sheep measles and hydatids, and the only way a producer may know if the condition is present is through post-mortem examination of the animals. The control of hydatids is of public health significance [34].Providing feedback to producers if the animals slaughtered from their farm were free from any defects and hence did not require any condemnation of offal or carcase/carcase meat. This is consistent with the findings from the review of the South Australian Enhanced Abattoir Surveillance Program [32].Expand the network of unbiased advisors that are currently available and accessible to sheep producers to discuss feedback and facilitate evidence-based management decisions at the enterprise level. These same advisors could assist with developing and implementing regional and jurisdictional animal health programs, including the monitoring, evaluation and review of feedback system.

To optimise the feedback system’s role in reducing pre-consumer losses, the system requires sustainable funding and resourcing mechanisms to support ongoing data collection, data analysis, data storage, timely distribution of accessible information, unbiased advisors, and monitoring of the system. With the benefits of reducing pre-consumer food losses extending beyond the farm gate and abattoir, there appears to be a greater role for government in these activities. This role would be one of supporting industry to achieve sustainability, similar to the South Australian Government’s role in the Enhanced Abattoir Surveillance Program [30], rather than a regulatory function. Government participation in programs to reduce food loss and waste increases the likelihood of program success [2].

With a global gap in official offal production data [8] and electronic individual-sheep identification becoming mandated in Australia [31], there is an opportunity for Australia to become a world leader in collecting and publishing these de-identified offal production data. Improving data collection and analysis will assist with improving the accuracy of food loss calculations in Australia [3] and in developing better informed policies on reducing food loss [35]. This shared information will assist sheep producers with management decisions at the enterprise level and benefit society more broadly through increased nutrient availability and increased sensitivity in natural resource management and assessments.

Further research with livestock agents, sheep meat processors, and government agencies is recommended to discuss their knowledge and attitudes to the feedback system. The feasibility of implementing the short-to-medium- and longer-term strategies to improve the Australian sheep meat value chain feedback system and reduce pre-consumer food losses that were developed because of the sheep producers; surveys and focus group discussions should also be discussed with the other value chain actors.

## Figures and Tables

**Figure 1 foods-12-04264-f001:**
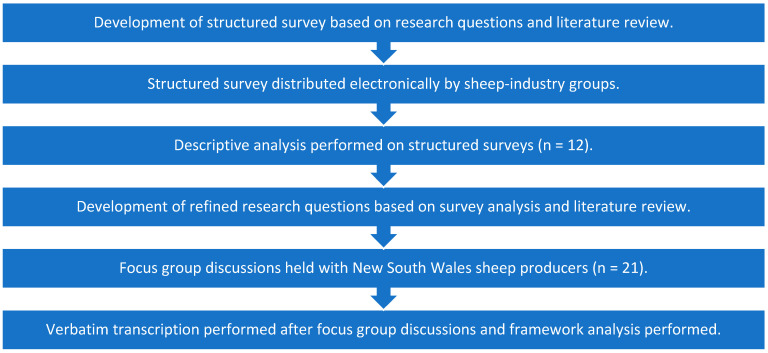
Outline of the study design used to explore Australian sheep producers’ attitude to post-harvest feedback.

**Figure 2 foods-12-04264-f002:**
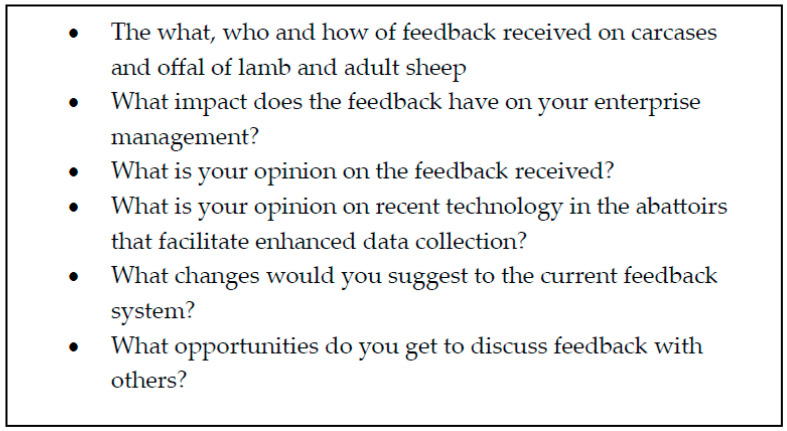
Semi-structured interview guide used to facilitate focus group discussions.

**Figure 3 foods-12-04264-f003:**
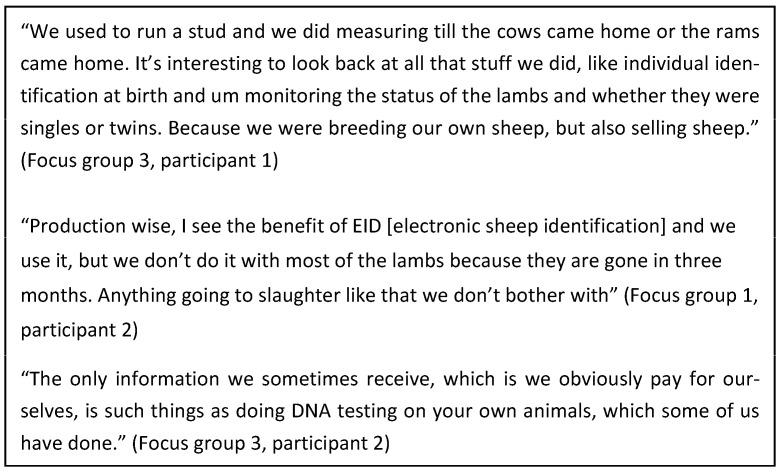
Quotes from participants on gathering information on breeding animals to improve enterprise productivity.

**Figure 4 foods-12-04264-f004:**
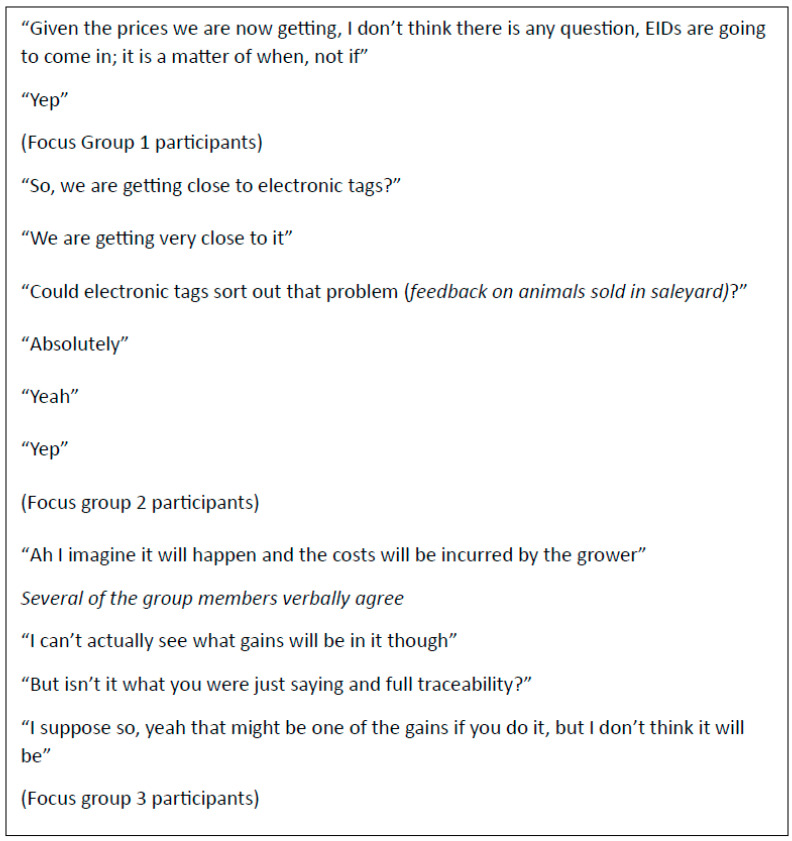
Quotes from group members when asked about the role of electronic individual-sheep identification in the Australian sheep meat value chain.

**Figure 5 foods-12-04264-f005:**
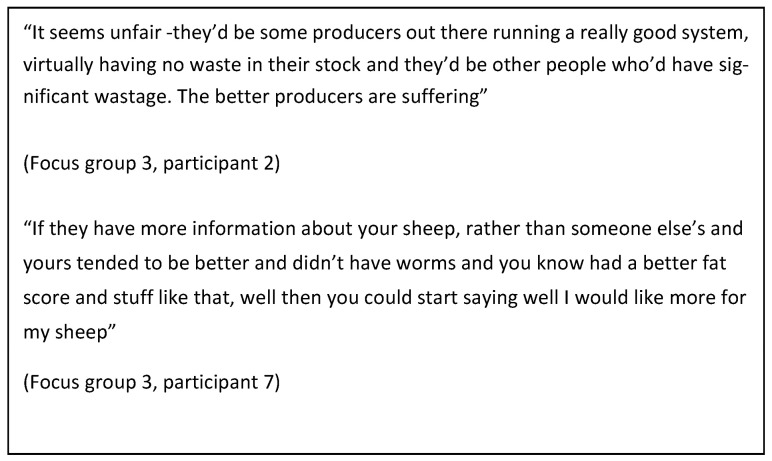
Quotes from participants in focus group 3 regarding the inequity of distributing the cost of downgraded offal and carcase meat across the industry.

**Figure 6 foods-12-04264-f006:**
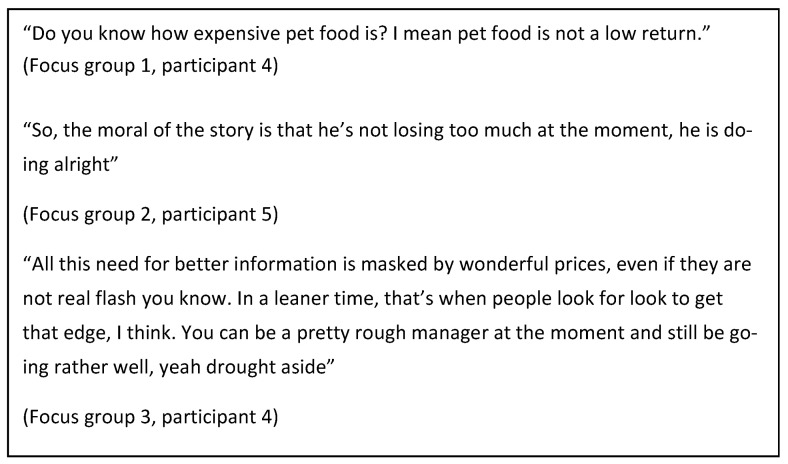
Quotes from participants of each group on the influence of the economy on producers and processors attitude towards feedback in the Australian sheep meat value chain.

**Table 1 foods-12-04264-t001:** Categories and their descriptions and codes that were developed during the framework analysis of the transcripts from the focus group discussions with New South Wales sheep producers regarding the feedback system.

Category Name	Description	Codes	Themes
Producer insights	Content of feedback received and process of how this is received both historically and in the present; actions members have taken based on the feedback and opportunities to discuss feedback	Feedback historical Feedback content current Feedback process current Feedback actions taken Opportunities to discuss feedback	Situation, knowledge, and attitudes of producers to the feedback system
Variation	Factors identified that create variation in the feedback received by New South Wales sheep producers	Enterprise type Abattoir/agent Sale method Jurisdiction	
Knowledge—industry	Producers’ knowledge of industry processes and practices and how this relates to feedback	Meat processing Industry systems Engaged producers Trusted sources of knowledge Knowledge is power	
Relationships	Stakeholder relationships and how they influence/impact the feedback system	Relationship with agent Relationship with abattoirs Relationship with government Role of other value chain actors Perceived power of producers	Barriers and enablers in the feedback system
Progress	Opinion of feedback and how it has changed over time, and opportunities into the future to make changes to the feedback system	Abattoir technology Opinion on feedback Future feedback suggestions Future actions from feedback	
Markets	The influence of markets on value of products and in turn the cost: benefit of a feedback system to different actors in the value chain	Branding and marketing Export carcase markets Potential offal markets	Equity in the feedback system
Value chain	Value chain factors that directly affect the feedback process in the Australian sheep meat value chain	Data governance Product ownership and value Sale and sale method Traceability Who pays for the feedback system?	
Planetary Health	The influence of positive human health, environment health and animal welfare outcomes on the drive for feedback in the Australian sheep meat value chain	Environment Human health Social license to farm	Sustainability of the value chain
Knowledge—food	How food technology, choices, and acceptance influence demand for products	Food acceptance Food choice Food knowledge Food technology	

## Data Availability

The data from this study are not available due to ethical reasons.

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
