# Peer review of "Australian Sheep Producers’ Knowledge of and Attitude towards Post-Harvest Feedback: A Mixed-Methods Case Study"

_foods, 2023, doi:10.3390/foods12234264_

Round 1

Reviewer 1 Report

Comments and Suggestions for Authors

Reducing food loss is vital for various reasons, as it has significant economic, environmental, and social impacts. This paper discusses an admirable and insightful study on Australian sheep producers' understanding and attitude towards post-harvest feedback and how they can affect pre-consumption losses in the Australian sheep meat value chain.

The article is written in a clear manner, without any spelling, grammar or punctuation errors. It is well organized and supported by a coherent data collection approach. Identifying key strategies to achieve continuous improvement to reduce pre-consumer losses is a very positive aspect of the work, which can impact the Australian sheep meat value chain feedback system.

Some small suggestions for change

L228 properties also identified “change with” properties were also identified

L436 system, although there are some challenges associated with “change with” However, some challenges are associated with

L496 being profitable markets for abattoirs, including diversion of offal “change with” being a profitable abattoir market, including diversifying offal

Please check the references section for consistency

L666 Animal production science “change with” Animal Production Science

L695 Environ Sci Technol “change with” Environmental Science & Technology

Reviewer 2 Report

Comments and Suggestions for Authors

The manuscript presents studies carried out to evaluate the knowledge and attitudes of sheep producers regarding the feedback system. In my opinion, some changes should be made to it.

The manuscript is long, so I suggest reducing the length of some sections like the introduction and results.

Figure 4 must also be included. Although it appears cited in the text, it does not appear in the manuscript.

The authors indicate that the structured survey obtained very few responses, which is why a descriptive analysis was carried out. Furthermore, the focus groups were oriented to the NSW region. So, some doubts arise about the extent of the results.

“The NSW feedback system was reflective of the base feedback system in place nationally at the time” (line 159)”. is this a result of this study or from a previous one? Some references should be included to sustain this statement. I consider it necessary to clarify this point because it indicates the representativeness of the study.

“The results of this study show that Australian sheep producers would like more detailed feedback about the products derived from their sheep” (line 514). It is not clear from the analysis described in the manuscript that the results can be extended to Australian producers in general. The authors are requested to clarify/deepen this point.

Reviewer 3 Report

Comments and Suggestions for Authors

The paper titled: "Australian sheep producers’ knowledge and attitude to Post-harvest feedback: a mixed methods case study" aims to determine the level of feedback given to sheep producers on the value of carcasses and offal. By determining this level, the authors suggest some short- and long-term strategies for implementation by the Australian government and extension services.

The authors used interesting methods to accomplish the tasks of the study, producing very insightful results that can be used by the Australian government and/or extension services, especially in the case of focus groups. However, there are some parts of the manuscript that I feel are too long and unnecessary or need to be explained more thoroughly. More detailed comments can be found in the attached file.

Reviewer 4 Report

Comments and Suggestions for Authors

Dear Authors,

Dear Editor,

the manuscript entitled "Australian sheep producers’ knowledge and attitude to post-harvest feedback: a mixed methods case study" have valuable insights to the field of sustainable livestock production and highlights practical steps that can be taken to improve the efficiency and sustainability of the Australian sheep meat industry. It addresses a specific industry need and offers actionable recommendations for positive change, making it both important and innovative.

The Introduction is well descibed. Also the Material and methods are well described. The questions in the manuscript are well chosen and can generally be applied to other survey studies (line 184-185). The questions are universal, which makes the manuscript universal, even though it describes research about Australia and sheep farms. This is important!

The discussion is also well written. The Authors give short, medium and long stategies. It is added value!

References - please add page number for example - point 5, 7, 11 ...

  Comments on the Quality of English Language

I think quality of English language is ok.
